

**Estimation of CFC-11 emissions from coal combustion in China**
Zhenzhen Niu [1], Shaofei Kong [1,2][*], Qin Yan [1], Yi Cheng [1], Huang Zheng [1], Yao Hu [1],
Jian Wu [1,2], Xujing Qin [1], Haoyu Dong [1], Weisi Jiang [1], Yingying Yan [1], Wei Liu [3],
Feng Ding [3], Yongqing Bai [4], Shihua Qi [1,2]
[1] Department of Atmospheric Sciences, School of Environmental Studies, China
University of Geosciences (Wuhan), Wuhan, 430078, China
[2] Research Centre for Complex Air Pollution of Hubei Province, Wuhan, 430078,
China
[3] Hubei Province Academy of Eco-Environmental Sciences, Wuhan, 430072, China
[4] Institute of Heavy Rain, China Meteorological Administration, Wuhan, 430205,
China
*Corresponding to Shaofei Kong (kongshaofei@cug.edu.cn)*





**Abstract**
The trichlorofluoromethane (CFC-11) emission from its production and use
(PAU) has drawn wide attention, while its combustion sources have been overlooked.
This study identified CFC-11 emission factors (EFs) as 3.6, 3.2, and 0.025 mg kg$^{-1}$
from the combustion of domestic chunk coal, honeycomb briquette, and coal-fired
power plant, respectively. A multi-year (2000~2021) emission inventory of CFC-11
from coal combustion was established in China. Results indicated that its annual
emission averaged 233.5 t yr$^{-1}$. It exhibited fluctuations and held an overall upward
trend, increasing from accounting for 0.8% of PAU emissions in 2000 to 9.8% in 2021,
with the peak value appearing in 2016. In Shandong and Hebei provinces with high
coal consumption amounts, the CFC-11 emissions from coal combustion increased by
approximately 74% during 2014~2017 compared to 2011~2012. At the Gosan station
close to Chinese mainland, CFC-11 emitted from coal combustion in Hebei and
Shandong was approximately occupied by ~30% of its average concentration during
January 2016. An additional climate effect of the clean heating and coal-to-electricity
policies in China was also observed, with an obvious decrease (2.2×10$^6$ t and 3.4×10$^7$
t) of CO$_2$-equivalent emission. This study provides substantial evidence of CFC-11
emission from coal combustion and highlights the role of combustion emission under
the background of reducing CFC-11 from PAU. The data compiled in this work can
found at https://doi.org/10.6084/m9.figshare.28523063 (Niu et al., 2025).

**Keywords:** Coal combustion; Trichlorofluoromethane emissions; Emission inventory;
Global warming potential; WRF-FLEXPART simulation



## 1. Introduction

Trichlorofluoromethane (CFC-11) has been widely used as a blowing agent for foams incorporated into buildings and consumer products since the 1950s (McCulloch et al., 2001; Rigby et al., 2019). The lifetime of CFC-11 exceeds 50 years, allowing it to accumulate in the atmosphere (Guo et al., 2009; Lickley et al., 2021; Rigby et al., 2013). CFC-11 could deplete stratospheric ozone through photodissociation (Fleming et al., 2020; Molina & Rowland, 1974), and served as a reference compound for calculating ozone depletion potential (ODP) (Western et al., 2023). Over a 100-year time horizon, CFC-11 has a global warming potential (GWP) thousands of times greater than that of carbon dioxide ($CO_2$) (Chiodo & Polvani, 2022; Polvani et al., 2020). An accurate understanding of CFC-11 emissions was helpful for assessing the impact of China's implementation of the Montreal Protocol (Fang et al., 2018).

The production and consumption of CFC-11 for emissive applications were phased out globally in 2010 according to the Montreal Protocol released in 1987 (Park et al., 2021). It results in a declining trend of its global atmospheric concentration (Park et al., 2021), and an expectation for ozone-layer recovery throughout the 21st century (Scientific assessment of ozone depletion, 2019). However, since 2012, the decline rate of atmospheric CFC-11 emission has significantly slowed by about 50% (Montzka et al., 2018). Eastern China has been identified as a hotspot for unexpected increased emissions of CFC-11 (Park et al., 2021). Former studies attributed it to new productions of CFC-11 in Eastern China (Montzka et al., 2018; Rigby et al., 2019), especially for blowing closed-cell insulating foam (McCulloch et al., 2001). Based on ambient monitoring data, former studies simulated CFC-11 emissions increased by 29.4% globally (Montzka et al., 2021), 58.3% in East Asia (Adcock et al., 2020), and 130.7% in eastern China (Park et al., 2021) during 2014~2017 compared to corresponding values of 2011~2012. They concluded that the annual emissions from existing CFC-11 banks alone could not fully explain the observed increase, highlighting a need to evaluate other potential sources for unexpected emissions



(Montzka et al., 2018). Therefore, the identification of new CFC-11 emission sources
and updating its emission estimation are urgent works.
There are two popular methods frequently adopted to estimate CFC-11 emissions.
The first was estimating its emissions based on atmospheric observation dataset,
including the inverse modeling approach which identified the CFC-11 emission using
two backward-running Lagrangian models, the UK Met Office Numerical
Atmospheric-dispersion Modelling Environment (NAME) and the FLEXible
PARTicle dispersion model (FLEXPART) (Park et al., 2021), and ratio method which
according to a correlation of CFC-11 with tracers holding clear emissions (Zhang et
al., 2014). Many tracers were adopted in former studies, such as carbon monoxide
(CO), chloroform ($CHCl_3$), and carbon tetrachloride ($CCl_4$) (Adcock et al., 2020;
Huang et al., 2021). CO was widely selected as its emission inventory was established
well and updated frequently by MEIC (Multiresolution Emission Inventory for China,
http://meicmodel.org.cn/#firstPage). The essential precondition is that the CFC-11 and
CO sources were co-located (Dhomse et al., 2019; Huang et al., 2021; Kim et al.,
2010). However, CO is a tracer for incomplete combustion (Zeng et al., 2020). If the
CFC-11 emission amounts were obtained by multiplying CO emission amounts with a
CFC-11/CO ratio from a linear fit, the results were untenable for the following two
reasons: (1) CFC-11/CO ratios selected varied in different researches, as 0.087
(Huang et al., 2021), 0.079 (Huang et al., 2021), 0.027~0.069 (Palmer et al., 2003),
and 0.022 (Shao et al., 2011). There was no objective criterion for selecting the CFC-
11/CO ratios. (2) The hypothetical co-locations of CFC-11 and CO do not mean that
their sources are the same. The obtained CFC-11 emission amounts through this
method actually mean that CFC-11 is only related to combustion sources.
The second was a bottom-up method. The CFC-11 emission inventory was
estimated based on the reported CFC-11 production and use (PAU) amounts from
different sectors, including foam blowing, solvents, and refrigerators (Fang et al.,
2018; Wan et al., 2009; Zhao et al., 2011), and combustion sources were always not





included. Additionally, in the fields of source profiles of volatile organic compounds
(VOCs), CFC-11 has been frequently detected for various types of combustion
sources (Gong et al., 2019; SPECIATE Version 5.3; Sha et al., 2021; Sun et al., 2019)
The emitted mass concentrations or emission factors of CFC-11 from various
combustion sources have also widely reported, such as power plant (12.5 μg m$^{-3}$) (Shi
et al., 2015), gasoline and diesel vehicles (0.01~0.06 mg km$^{-1}$) (Wang et al., 2020),
and coal combustion (0.07~0.51 ppbv) (Li et al., 2003). CFC-11 can be formed by the
combustion of coal that contains the necessary elements of carbon, chlorine, and
fluorine (Jin et al., 2025; Luo et al., 2004). The level of CFC-11 has been detected at
ppb levels in combustion (Pons et al., 2019), which is 3 magnitudes higher than its
ambient levels. To the best of our knowledge, the emission inventory of CFC-11
emissions from combustion sources has not been reported.
To sum up, we detected the emission factors (EFs) of CFC-11 from domestic
coal combustion (chunk coal and honeycomb briquette) and coal-fired power plants
with a unified dilution sampling method. An emission inventory with high spatial
resolution of CFC-11 from coal combustion in China during 2000~2021 was first
established. The variation trends of CFC-11 emitted from coal combustion and PAU
were compared. The impact of CFC-11 emissions from coal combustion in the
hotspots of Shandong and Hebei provinces on coastal air was simulated with the
WRF-FLEXPART model. This study provides a quantitative assessment of CFC-11
emissions from coal combustion in China, which will provide new insights for
identifying its variation trend in ambient air and refining the projection of
stratospheric ozone layer recovery.
**2. Methods**
**2.1 Source sampling**
To ensure the representativeness and applicability of the emission factors, the
combustion experiments were designed to closely simulate real-world domestic coal
combustion conditions in rural China. A total of 10 kinds of chunk coals and 11 kinds



of honeycomb briquettes were collected and burned in our combustion lab in Wuhan.
The annual average ambient level of CFC-11 was 0.6 μg m$^{-3}$ in the year 2023. Fuels
were collected from eight agricultural regions of China, including the Northeast Plain
(Heilongjiang, Jilin, and Liaoning), Arid and semi-arid regions of north China (Inner
Mongolia, Ningxia, Gansu, and Xinjiang), Loess Plateau (Shaanxi and Shanxi), North
China plain (Anhui, Beijing, Hebei, Henan, Jiangsu, Shandong, Shanghai and Tianjin),
Yangtze Plain (Hubei, Hunan, Jiangxi, and Zhejiang), Sichuan Basin (Sichuan and
Chongqing), Yunnan-Guizhou Plateau (Guangxi, Guizhou, and Yunnan), Tibet Plateau
(Qinghai and Xizang) and South China (Fujian, Guangdong, and Hainan). The
specific information on fuel collection can be seen in Table S1. If the fuel in one
region had not been collected, the fuel emission characteristics of the neighboring
provinces were used as a substitute.
The stove used was a typical household furnace purchased from a local market,
with an outer diameter, inner diameter, and height of 30, 12, and 43 cm, respectively.
For each test, about 0.8 kg chunk coals and 1.5 kg honeycomb briquettes (three pieces)
were burned. The combustion process was manually operated to replicate the real
usage patterns of rural households. An electronic scale was positioned at the bottom of
the stove to record the variation of fuel quality. Flue gases were drawn with a
sampling gun (1.5 m higher than the flame) and then diluted ~30 times with a dilution
system (TH-150, Wuhan Tianhong Ltd., China). The equipment settings can be
referred to our previous studies (Yan et al., 2020, 2022). The diluted gases were
collected into a 4 L Tedlar bag at a flow rate of 150 mL min$^{-1}$. The specific sampling
systems can be seen in Figure S1. Each sampling practice covered a whole fuel-
burning period. A total of 52 sets of samples were obtained.
For coal-fired power plants, 6 L summa cans were used to collect the flue gas
after diluted. Each sampling time lasted for about 23 hours. The power plant has
adopted ultra-low emission pollutant control measures, including wet desulfurization,



electric dust precipitation, and denitrification. Detailed information on the plant and
field sampling settings can be found in our former research (Zeng et al., 2021).
**2.2 CFC-11 analysis, quality assurance and quality control**
CFC-11 was analyzed by a gas chromatography/mass spectrometry (GC-MS,
Agilent 7820A/5977E). Samples were pretreated through a cold trap pre-concentrator
before into the BD-624 chromatographic columns (60 m × 0.25 mm × 1.4 μm). 300
mL gases were extracted from the Tedlar bags or summa cans into a cold trap to
remove water and $CO_2$. Then, the concentrated gases were sent to the gas
chromatography with helium gas as the carrier gas. The gases passed through the
chromatography column were divided into two sections, one for a FID detector and
another one for a MS detector. The chromatographic column temperature increased
from 35 ℃ to 180 ℃, at a rate of 6 ℃ $min^{-1}$. The temperature for both the FID and
MS detectors was 200 ℃. The EI ionization mode of mass spectrum was adopted. The
electron energy was 70 eV. An internal standard method was used to calculate the
concentration. Four internal standard substances including bromochloromethane, 1,4-
difluorobenzene, chlorobenzene, and 4-bromofluorobenzene are used. CFC-11 was
determined with a Mass Selective Detector (MSD) by the target ion at m/z 103/101,
and this method was widely used in previous research (Huang et al., 2021; Jin et al.,
2025; Zhang et al., 2014). GC-MS was also used in other research for CFC-11
observation (Park et al., 2021)
For quality control and quality assurance, tedlar bags were not reused in this
study. A system blank test was conducted, after every 10 samples were analyzed and
after the samples were analyzed at high concentrations. The calibration curves were
updated monthly. A parallel sample was analyzed for every 10 samples or each batch
(less than 10 samples), to ensure that the relative deviations of the targets were less
than or equal to 30%. If the relative deviations exceeded 30%, then the sample was re-
analyzed. Before each sample analysis, the air and water, the relative abundance of
water, nitrogen, and oxygen should be less than 10%, otherwise the leakage of the





instrument system should be checked. The detection limit of CFC-11 was 0.15 μg m$^{-3}$.
The concentration of CFC-11 in the blank sample of the instrument is 0.
**2.3 Calculation method of CFC-11 emission**
The EFs of CFC-11 from domestic coal combustion were calculated as follows:
$$EF_i = \frac{(c_i \times \frac{v}{v_1} \times n - c_0) \times v_1 \times t \times 10^{-6}}{M_i} \qquad (1)$$

Where $i$ stood for the fuel type; $EFi$ was the CFC-11 emission factor for
combustion of fuel $i$, mg kg$^{-1}$; $c_i$ was the mass concentration of CFC-11 in the
sampling port after the combustion of fuel $i$, μg m$^{-3}$; $v$ indicated the flow rate of flue
gas, L min$^{-1}$; $v_i$ indicated the sampling flow rate, L min$^{-1}$; $n$ stood for dilution ratio; $c_0$
was the mass concentration of CFC-11 in the atmospheric environment, in this study
was 0.6 μg m$^{-3}$; $t$ was the sampling time, min; $M_i$ was the weight of fuel $i$ burned, kg.
The average mass concentration of CFC-11 from domestic coal combustion was 93.9
±90.4 μg m$^{-3}$, which was 150.7 times that of the ambient concentration, which
indicated that the impact of ambient CFC-11 concentrations on its emission from coal
combustion sources can be ignored.
The EFs of CFC-11 from coal-fired power plants were calculated as follows:
$$m_i = c_i \times v_1 \times t \times 10^{-6} \qquad (2)$$

$$EF_{ij} = \frac{v \times m_i \times r_j^2 \times n}{v_1 \times M_i \times r^2} \qquad (3)$$

Where $i$ stood for the fuel type; $m_i$ was the emission amount of CFC-11 released
from the combustion of fuel $i$, mg; $c_i$ was the mass concentration of CFC-11 from
stack, which ignored the CFC-11 ambient concentration, μg m$^{-3}$; $v_1$ indicated the
sampling flow rate, L min$^{-1}$; $t$ was the sampling time, min; $EF_{ij}$ was the CFC-11
emission factor emitted by the combustion of coal $i$ from power plant $j$, mg kg$^{-1}$; $r_j$
was the semidiameter of the stack at the sampling point, m; $r$ was the semidiameter of
the sampling nozzle, m; $n$ stood for dilution ratio; $v$ indicated the flow rate of flue gas,
L min$^{-1}$; $M_i$ was the weight of coal $i$ burned, kg.
The CFC-11 emission amounts were calculated by multiplying its EFs (Table S2)
with corresponding coal consumption amounts each year in China. The coal



consumption amounts for each province of China from 2000~2021 were obtained
from the China Energy Statistical Yearbook, and there were no data for Hong Kong,
Macao, Taiwan, and Xizang (China energy statistical yearbook). The coal
consumption amounts from 2022~2060 were calculated according to the decrease rate
in references (Wu et al., 2024; Energy Foundation, 2024). The spatial distribution of
CFC-11 from domestic coal combustion was allocated according to the 2000~2018
land use data with 30 m*30 m and 2000~2021 population distribution data with 1
km*1 km (WorldPop and Center for International Earth Science Information Network)
(Gong et al., 2019, 2020). The land use data for 2019~2021 used the data in 2018. The
Point of Interest (POI) data of industrial were obtained to allocate the CFC-11
emission from coal-fired power plants into each plant (Figure S2). The specific
calculation and allocation method could be found in our former studies (Cheng et al.,
2022; Wu et al., 2021).

The $CO_2$-equivalent ($CO_2$-eq) emissions were calculated by multiplying the

CFC-11 emission amounts with its global warming potential (GWP) value of 7090
(Burkholder et al., 2022).

**222 2.4 WRF-FLEXPART modeling**

Previous studies identified Shandong and Hebei provinces as the dominant

source regions for CFC-11 detected on islands near Korea and Japan (Park et al.,
2021). Here, we tried to explore the influence of CFC-11 emissions from coal
combustion in the two provinces on its ambient levels. FLEXPART was usually
employed for inverse estimating CFC-11 emissions by former researchers (An et al.,
2012; Park et al., 2021; Rigby et al., 2019). Here, January was heating period with
higher coal combustion, and the year 2016 had higher CFC-11 emissions (Montzka et
al., 2018), January 2016 was selected as the simulated period. The meteorological
input data were obtained and downloaded from National Centers for Environmental
Prediction (NCEP) Final Analysis (FNL; https://rda.ucar.edu/), which provided the
lateral boundary conditions and initial meteorological fields for the simulation. The



FNL data had a horizontal resolution of 1°× 1° and a temporal interval of 6 hours. The
simulation domain encompassed the East Asian region (within the boundary of
20~47°N and 110~140°E). Hebei and Shandong were identified as the primary CFC-
11 release areas. The simulation period was set for January 2016 (similar to the
monitoring period in former studies) (Park et al., 2021), utilizing the forward
modeling approach for analysis. Air parcels were released from the gridded emission
areas over Hebei and Shandong provinces at altitudes from the surface up to 100 m,
reflecting the near-ground emissions from coal combustion.
**3. Results and discussion**
**3.1 CFC-11 EFs for coal combustion and comparison with other sources**
The EFs of CFC-11 from chunk coal and honeycomb briquette combustion
varied from 0.3~12.7 mg kg$^{-1}$ (3.6±2.9 mg kg$^{-1}$) and 0.6~4.2 mg kg$^{-1}$ (3.2±0.7 mg
kg$^{-1}$), respectively (Figure 1). These values were 144 and 128 times higher than the
EF for coal-fired power plant (0.025 mg kg$^{-1}$). The honeycomb briquette had higher
combustion efficiency than chunk coal, which may reduce the release of chloride and
formation of CFC-11 (Li et al., 2016). The much lower EFs for coal-fired power plant
could be related to the high combustion temperature and series of flue gas treatment
measures (Yan et al., 2016). CFC-11 includes chloride (Cl) and fluoride (F), the
formation of CFC-11 needs the participation of Cl and F. F and Cl were widely
distributed in Chinese coal. Former studies indicated that F content was 20~300
mg/kg from coals in the North China Plate and Northwest China, lower than the
Southwest China (50~3000 mg/kg) (Luo et al., 2004). The chlorine content of
bituminous coal was 252.5 mg kg$^{-1}$ in China (Jin et al., 2025). The formation and
emission mechanisms of CFC-11 during coal combustion remained unclear. There
was only one old literature reported that CFC-11 could be detected from the
combustion of all tested 23 types of coal, and the release of CFC-11 peaked at a
combustion temperature of 400 ℃ (Li et al., 2004). Coal combustion could emit
halogenated organic compounds, such as methyl chloride (CH$_3$Cl) and chloroform
($CH_2Cl_2$) (Liu et al., 2024). Recent research presented the possible formation route of
CFC-11 from above halogenated organic compounds in the iron and steel industry,
based on the traditional liquid-phase fluorination method (Liu et al., 2024). This
exploratory study primarily deduced that the formation conditions of CFC-11 in coal
combustion were similar to the industrial synthesis conditions of CFC-11. The
transformation pathways of solid fluoride in coal to CFC-11 and influencing
parameters were still a puzzle. The mechanisms driving the formation and release of
CFC-11, as well as the dominant influencing factors, remain unexplored and warrant
further investigation.

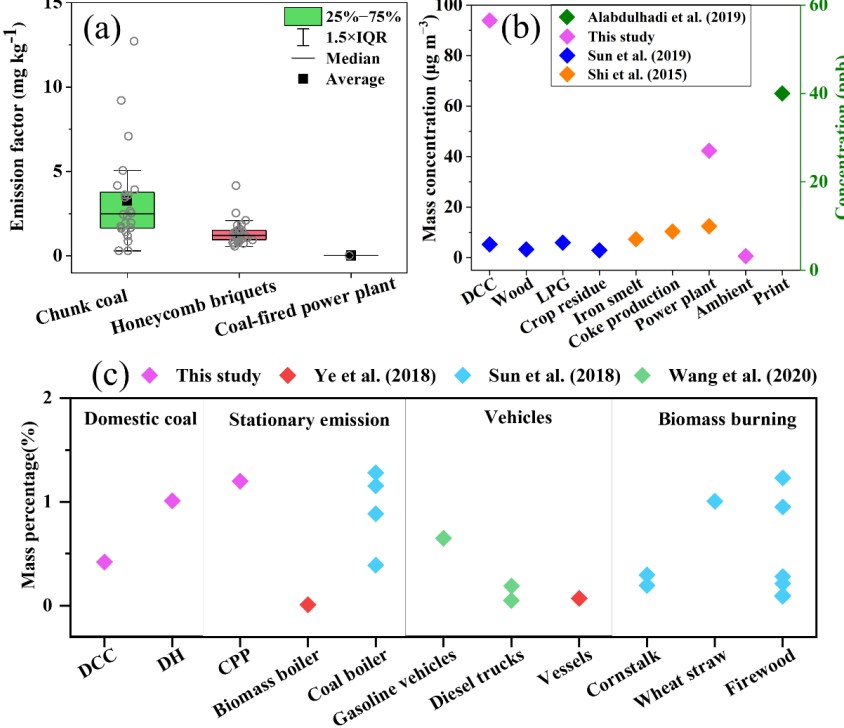


**Figure 1.** Comparison of CFC-11 emission from coal combustion and other sources for its
emission factor (a), mass concentration (b), and mass percentage (c) in total VOCs. The VOCs
included 102, 61, 107, 101, 98, and 102 species in this study, Sun et al. (2019), Shi et al. (2015),
Ye et al. (2018), Sun et al. (2018), and Wang et al. (2020), respectively. DCC means domestic
chunk coal, DH means domestic honeycomb, and CPP means coal-fired power plant.

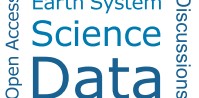

Previous studies have reported the emission of CFC-11 from other anthropogenic

sources (Gong et al., 2019; Sun et al., 2019). Domestic anthracite coal combustion
(5.2 µg m$^{-3}$) (Sun et al., 2019), coating (44.5 µg m$^{-3}$ and 91.0 µg m$^{-3}$), and printing
(40 ppb and 10.9 µg m$^{-3}$) all emitted CFC-11 (Alabdulhadi et al., 2019; Shen et al.,
2018). Figure 1b presented the CFC-11 mass concentration for reported combustion
sources in the literature, coal-fired power plants (42.3 µg m$^{-3}$), iron smelting (7.3 µg
m$^{-3}$), coke production (10.3 µg m$^{-3}$), and coal-fired power plants (12.5 µg m$^{-3}$) (Shi et
al., 2015) all were the CFC-11 emission sources. The CFC-11 accounted for 0.4%,
1.0%, and 1.2% of the total volatile organic compounds (VOCs) detected from the
combustion of chunk coal, honeycomb briquette, and coal-fired power plant in this
study, respectively. These proportions were higher than those for stationary
combustion of biomass (0.01%) (Ye, 2018), heavy-duty diesel trucks (0.05% or 0.2%)
(Wang et al., 2020), vessels (0.07%) (Ye, 2018) and corn stover burning (0.3% or
0.2%) reported in the literature (Figure 1c) (Sun et al., 2018). The CFC-11 mass
percentages for VOCs emitted from stationary coal combustion and firewood burning
were 0.4%~1.3% and 0.1%~1.2% (Sun et al., 2018), similar to this study. Ground
measurement campaigns also recorded high CFC-11 levels from specific events, such
as 626 ppt and 658 ppt for garbage burning and a landfill fire near Mecca,
respectively (Simpson et al., 2022). Although combustion-related CFC-11 emissions
were influenced by combustion conditions, these findings provided evidence for the
contribution of coal combustion and other combustion sources to overall CFC-11
emissions.
**3.2 Spatial-temporal distribution of CFC-11 from coal combustion in China**

The annual CFC-11 emissions from coal combustion in China during 2000~2021

exhibited fluctuations and an overall upward trend, peaking at 268.7 t yr$^{-1}$ in 2016
(Figures 2a−2b). CFC-11 emissions increased after 2012, consistent with previous
studies that reported rising CFC-11 concentrations in ambient air (Adcock et al., 2020;
Montzka et al., 2018; Rigby et al., 2019). Approximately 40%~60% of the global
increase in CFC concentration was attributed to China, particularly Shandong and
Hebei provinces (Adcock et al., 2020; Montzka et al., 2018; Rigby et al., 2019). This
study found marked increases in CFC-11 emissions from Hebei (14.3 t yr⁻¹) and
Shandong (11.0 t yr⁻¹) in 2013, which were 2.2 and 1.4 times the respective emission
amounts in 2012 (Figure 3). The emission of CFC-11 emissions from coal combustion
in China fluctuated during 2001~2021, averaging 233.5 t yr⁻¹. The contribution of
coal combustion to CFC-11 emissions on a global scale needs further research.

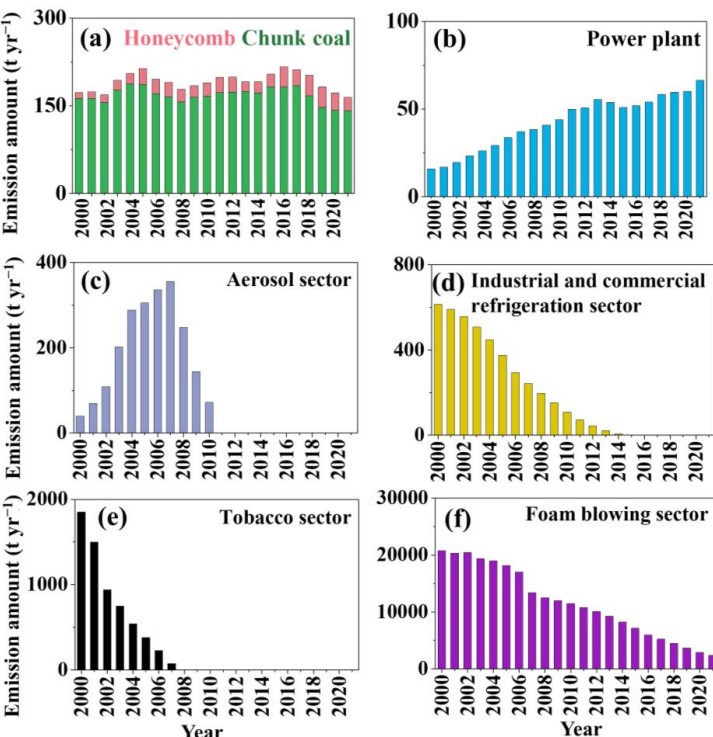


**Figure 2.** Annual CFC-11 emission from domestic coal combustion (a), coal-fired power plant (b),
aerosol sectors (c), industrial and commercial refrigeration sector (d), tobacco sector (e), and foam
blowing sector (f) in China. The data for (a) and (b) were calculated in this study. The data for
(c)~(f) referred to Fang et al. (2018).

Although the contribution of domestic chunk coal combustion to CFC-11 annual

emissions decreased from 2000 to 2021, it was still the dominant contributor,
accounting for 60.9%~86.4% of CFC-11 emissions of domestic coal combustion in



China (Figure 3). By 2021, the cumulative CFC-11 emissions from coal combustion
reached 5135.7 t in China (Figure S3), with domestic coal combustion contributing
4200.0 t and coal-fired power plant contributing 935.7 t. With the transformation of
China's energy structure, the proportion of CFC-11 emissions from coal-fired power
plants in total CFC-11 emissions from coal combustion increased from 7.9% in 2000
to 18.2% in 2021.

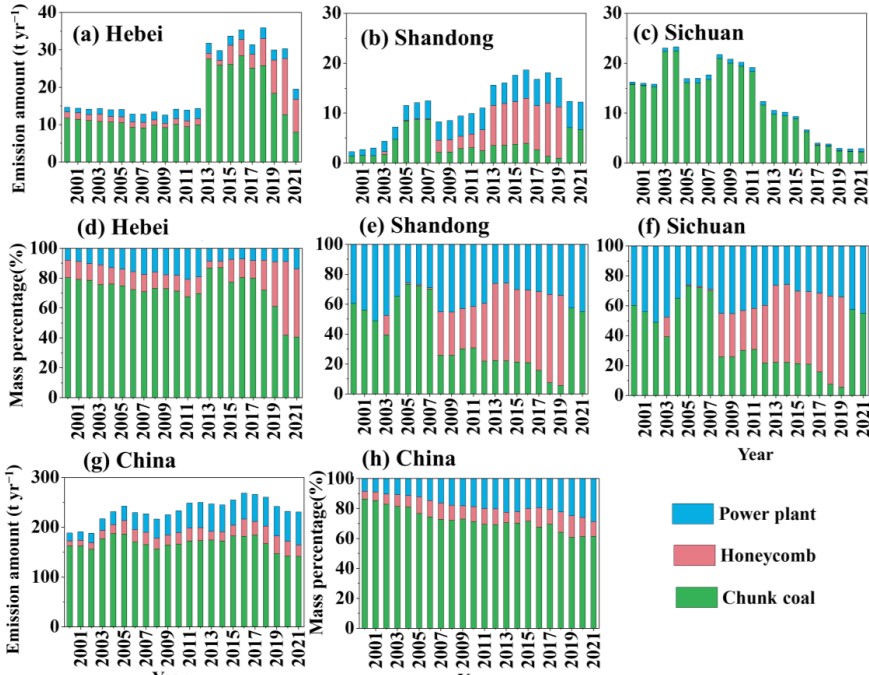


**Figure 3.** The CFC-11 emissions (a~c) and mass percentages (d~f) from power plant, domestic

chunk coal, and honeycomb combustion in Hebei, Shandong, and Sichuan provinces.

Previous studies have constructed CFC-11 emission inventories for its PAU

processes and formed a PAU emission bank (Fang et al., 2018; Wan et al., 2009),
which mainly included the sectors of aerosol, industrial and commercial refrigeration,
tobacco, and foam-blowing in China as Figures 2c−2f shown. The CFC-11 emission
amounts from coal combustion were comparable with those from aerosol sector and
industrial and commercial refrigeration. The CFC-11 emissions from aerosol sector
and tobacco sector disappeared after 2010 and 2007 as the Montreal Protocol,



respectively. After 2015, the foam-blowing sector became the sole contributor to
CFC-11 emissions among these sectors, with its emission declining to 7155.9 t yr$^{-1}$. If
all other CFC-11 emissions from PAU sources were gradually getting to zero, while
the CFC-11 emissions from coal combustion persisted, the influence of CFC-11
emissions from coal combustion should be considered at that time, especially when
the CFC-11 emissions from PAU were cleared to zero.
Figure S4 presents the CFC-11 emissions from coal combustion in different
provinces in China during 2000~2021. Provinces in heating areas exhibited high
CFC-11 emissions throughout the study period, they were in the north of China. Such
as Inner Mongolia, Hebei, Henan, Xinjiang, Shandong, and Shanxi, the CFC-11
emissions in 2021 were 36.1 t, 19.5 t, 10.0 t, 21.2 t, 12.2 t, and 15.2 t respectively.
Figure 4 shows the CFC-11 emission intensity of CFC-11 from domestic coal
combustion. High-emission areas were consistently concentrated in the North China
Plain, including Hebei, Shandong, and Henan, where residential coal consumption has
historically been significant. Over time, these high-emission zones became more
pronounced, particularly after 2013.

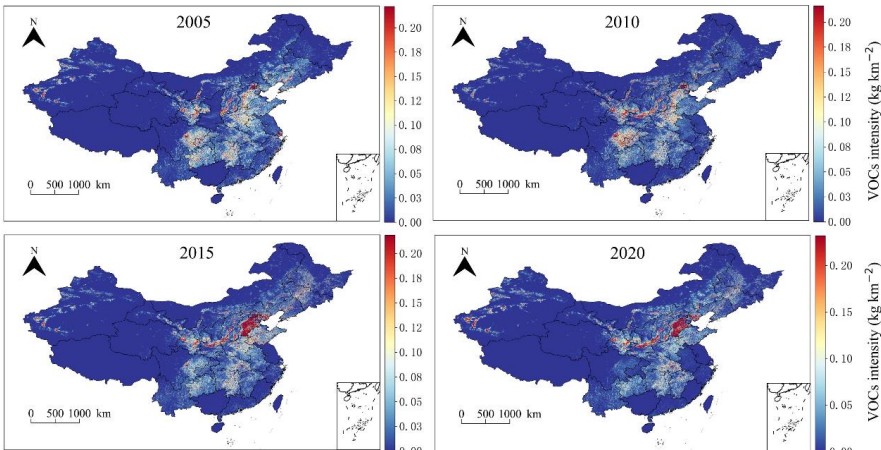

**Figure 4.** CFC-11 emission intensity of CFC-11 from domestic coal combustion in 2005, 2010,
2015, and 2020.

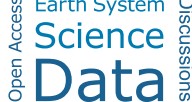

### 3.3 Comparison with CFC-11 emission obtained from CFC/CO ratios

In this study, the slope of CFC-11 and CO from coal combustion was 0.447 (Figure 5), higher than 0.039~0.087 in the atmosphere (Huang et al., 2021). CFC-11 emission inventory from coal combustion was obtained according to the CO emission inventory from coal combustion and CFC-11/CO ratio. Figure 5b presents the CFC-11 emission from coal combustion using CO tracer method and bottom-up method. The CFC-11 emission through CO tracer method was 7872~60466 kt yr$^{-1}$, much higher than the emission using bottom-up method (7.9~60.8 t yr$^{-1}$). Although the ratio method using CO as a tracer was commonly applied in estimations, it might lead to an overestimation of CFC-11 emissions. Since CO had many emission sources, if the ratio was calculated using CFC-11/CO in the atmospheric concentration, then CFC-11 was also assumed to come from these emission sources. From previous research, CFC-11 emission from combustion sources, including industrial processes, vehicle emissions, garbage burning, LPG, and biomass burning, had long been overlooked (Shen et al., 2018; Wu & Xie, 2017; Zhang et al., 2020). However, the growing significance of these emissions highlighted the need for a more comprehensive evaluation of all potential sources for CFC-11, including above combustion sources and non-combustion sources like fuel oil storage, oil transportation, and printing facilities (Alabdulhadi et al., 2019b).

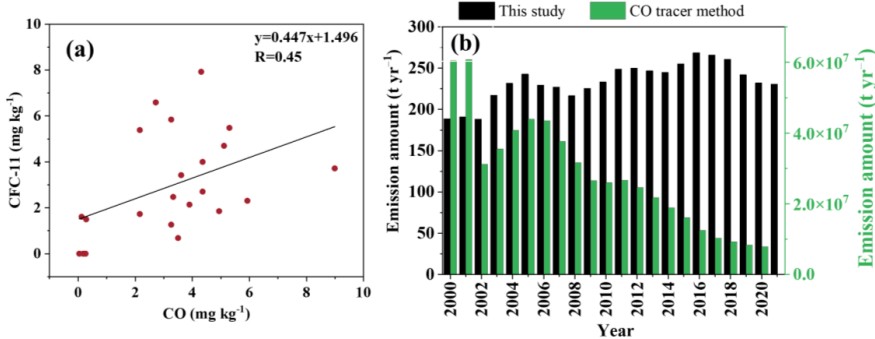

**Figure 5.** Interspecies correlations of CFC-11 with CO from coal combustion, R=Pearson's r (a), and comparison of CFC-11 emissions from coal combustion between this study and the CO tracer



method. The emission inventory of CO for coal combustion was refereed to previous studies (b)
(Liu et al., 2015; Peng et al., 2019; Tong et al., 2018).

**3.4 Increasing importance of coal combustion in CFC-11 emission**

A lot of research calculated the CFC-11 emissions in China and even globally
based on ambient monitoring data (Table S3), the CFC-11 emissions from coal
combustion were smaller than all CFC-11 emissions in China. The proportion of
CFC-11 emissions from coal combustion relative to its bank emissions increased year
by year from 2000 (Figure 6a). The annual CFC-11 emissions from coal combustion
in China from 2000 to 2021 varied from 188.5~268.7 t yr$^{-1}$, accounting for 1.5%~2.1%
of the global increase in CFC-11 emission of 13±5 kt yr$^{-1}$ reported in the literature
(Montzka et al., 2018). In 2000, the CFC-11 emission from coal combustion was
188.5 t yr$^{-1}$, only accounting for 0.8% of PAU emissions in China. By 2021, however,
CFC-11 emissions from coal combustion had risen to 9.8% of PAU emissions
according to Figure 6a. After 2025, the CFC-11 emissions from PAU in China was 0,
but the CFC-11 from coal combustion still existed as seen in Figure S5, subsequent
controls should give greater consideration to coal combustion because of its
widespread sources (Jin et al., 2025).

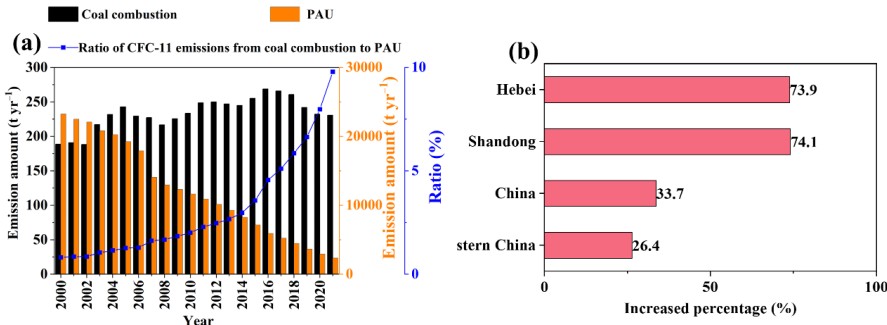


**Figure 6.** (a) Comparison of CFC-11 emission amounts from coal combustion (including coal

consumed in domestic use and power plant) with CFC-11 emission from production and use (PAU)
including aerosol sectors, tobacco sector, foam blowing sector, and industrial and commercial
refrigeration sector in China (Fang et al., 2018). (b) The increased percentages for CFC-11 in this
study in 2014~2017 compared to 2011~2012.



CFC-11 emissions from coal combustion increased sharply in 2013 as seen in

Figure 2. From 2014 to 2017, CFC-11 emissions from coal combustion in China,
Shandong Province, and Hebei Province increased compared to their corresponding
emissions in the 2011~2012 period, the increasing ratios were 33.7%, 74.1%, and
73.9%, respectively (Figure 6b). Previous studies indicated that the concentration of
CFC-11 in the northern hemisphere's atmosphere in 1995 was approximately 267 ppt
(Montzka et al., 2018). In 2060, the concentration of CFC-11 is 136.7 ppt, and in 2100,
the concentration of CFC-11 still has 69.5 ppt (Daniel et al., 2022). Considering the
lifetime of CFC-11 is about 52 years (Burkholder et al., 2022), we inferred that coal
combustion might slightly contribute to the 136.7 ppt and 69.5 ppt of CFC-11.
**3.5 Additional climate benefits of clean heating and coal-to-electricity policies**

As the clean coal and heating policies were implemented, the coal consumption

structures differed in southern (taking Sichuan as an example) and northern provinces
(Hebei and Shandong) and changed quickly, which led to a clear variation of CFC-11
emission from coal combustion (Figure 3). The CFC-11 emission from honeycomb
briquette combustion increased for Hebei and Shandong Province after 2013 when the
Action Plan for Air Pollution Prevention and Control in China was released (Geng et
al., 2024). In Sichuan province, CFC-11 emissions from chunk coal combustion
decreased significantly, especially after 2013. By 2020, no CFC-11 emissions from
honeycomb briquette combustion were detected in Shandong and Sichuan Province.

With the replacement of chunk coal with honeycomb briquette and coal-to-

electricity policy, the emission of CFC-11 from chunk coal combustion gradually
decreased after 2016 for China (Figure S5). Domestic coal combustion was projected
to cease entirely by 2030 (Energy Foundation, 2024), and CFC-11 emissions from
coal-fired power plant decreased gradually to zero in 2060 to realize carbon neutrality
(Figure S5) (Wu et al., 2024). From 2000 to 2060, the cumulative CFC-11 emissions
from coal combustion in China will be 7115.0 t. Even though the coal consumption
structure had changed (Shen et al., 2022), coal combustion remained a stable emission



source of CFC-11. Its accumulated emission amounts were similar to the historical
(2000~2060) CFC-11 emissions from tobacco sector (6263 t), and higher than that of
aerosol sector (4233 t) and industrial and commercial refrigeration (2169 t).
Figure 7 illustrates the $CO_2$-eq emissions in China from coal combustion
between 2000 to 2021. In 2021, the $CO_2$-eq emissions reached $1.7 \times 10^6$ t $yr^{-1}$,
accounting for 0.02% of total anthropogenic $CO_2$ emissions in China and 0.2% of $CO_2$
emissions from cement from Global Carbon Atlas. This value accounted for 0.03% of
China's forest carbon sink ($6.6 \times 10^9$ t $CO_2$) (Liu et al., 2015; Pan et al., 2011). These
findings highlighted the need to reassess the role of CFC-11 from combustion
emissions in global warming potential. From Figure 7b, the contribution of chunk coal
combustion to $CO_2$-eq emissions decreased from 89.3% in 2000 to 63.3% in 2021 and
would decrease to 0 after 2030. The replacement of chunk coal with honeycomb
briquette resulted in a decrease of 25.2% in chunk coal usage and 8.9% in honeycomb
usage (China Energy Statistical Yearbook). During 2000~2021, if all chunk coal was
replaced by honeycomb briquette, CFC-11 and $CO_2$-eq emissions would be reduced
by 10.6~16.0 t $yr^{-1}$ and $7.5 \times 10^4$~$1.1 \times 10^5$ t $yr^{-1}$, respectively (Figures S6 and 7c).
This study verified the necessity of energy mix adjustment and the use of clean energy,
from the aspect of co-prevention and control of multi-pollutants and the win-win in
climate-environmental benefits (Shen et al., 2019; Shen et al., 2021; Tao et al., 2021).
Although the decreased CFC-11 and $CO_2$-eq emissions were small, their impacts on
the ozone layer and climate change should not be underestimated because of the
extensive and universal sources of CFC-11 emissions (Jin et al., 2025).
In contrast, the contribution of coal-fired power plant to $CO_2$-eq emissions
increased from 6.8% in 2000 to 28.0% in 2021 and was expected to rise to 100% after
2030. The coal-to-electricity strategy implemented in China increased the coal
consumption in power plant (Wang et al., 2020), significantly reducing CFC-11
emissions from domestic coal combustion by 170.1~252.0 t $yr^{-1}$ and reducing $CO_2$-eq
emissions by $1.2 \times 10^6$~$1.8 \times 10^6$ t $yr^{-1}$ during 2000~2021 (Figures S6 and 5d). It



indicates that transitioning to cleaner coal alternatives can not only improve the air
quality in the North China Plain (Fang et al., 2019), but also yield unexpected
significant climate benefits by reducing CFC-11 and $CO_2$-eq emissions. However,
CFC-11 has a very large and uncertain indirect radiative cooling effect due to its
depletion of Ozone, resulting in an indirect GWP of -4390 (Daniel et al., 2022). $CO_2$-
eq emission in this study was calculated using direct GWP, relying solely on the direct
GWP might overestimate its climate impact. Therefore, a more comprehensive
approach was essential for accurately assessing the full climate impact of CFC-11 and
informing effective mitigation strategies.

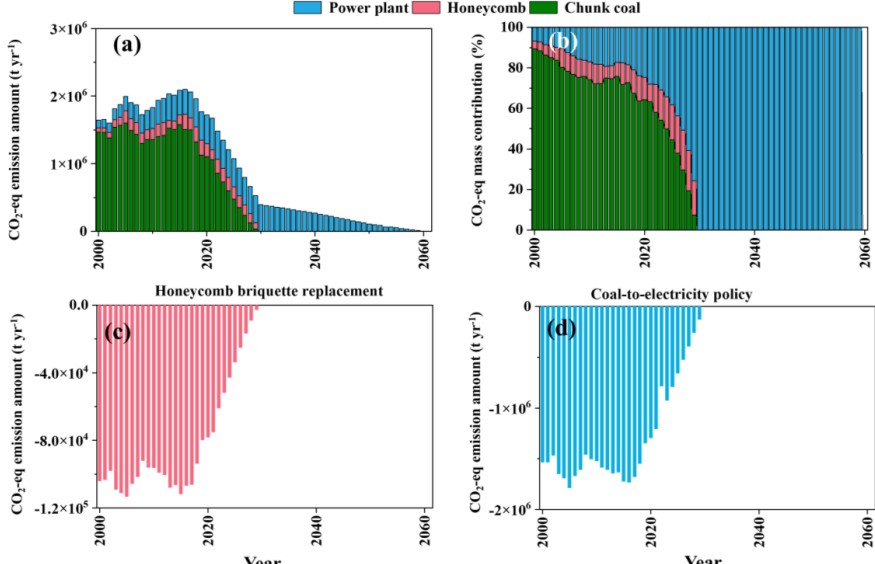


**Figure 7.** $CO_2$-eq emissions for CFC-11 emitted from coal combustion in China in this study (a)
and (b). The changes of $CO_2$-eq emissions from CFC-11, if coal was replaced by honeycomb (c)
and if domestic coal was replaced by electricity produced in coal-fired power plant (d).
**3.6 The influence of CFC-11 from coal combustion on ambient concentration**

Former researchers indicated that additional emission of CFC-11 was found in

2016 (Montzka et al., 2018; Rigby et al., 2019), monthly CFC-11 emission is
presented in Figure 8. The monthly CFC-11 emissions from domestic coal combustion
were allocated according to Wu et al. (2021), from coal-fired power plant were



allocated according to the power generation volume from National Bureau of
Statistics of China (https://www.stats.gov.cn/). The higher CFC-11 emissions from
domestic coal combustion were in cold months, January (22.1 t), February (22.1 t),
October (22.1 t), November (26.1 t), and December (24.1 t). The higher CFC-11
emissions from coal-fired power plant were in August (4.9 t) and December (5.0 t).

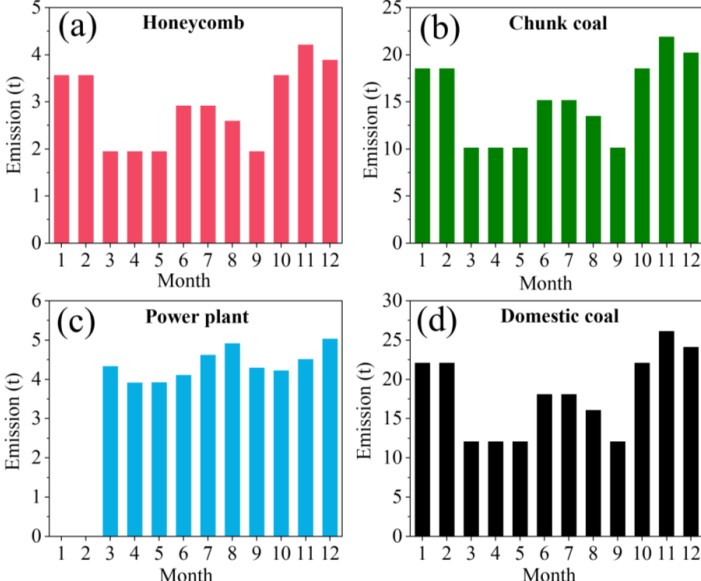


**Figure 8.** Monthly CFC-11 emissions from domestic honeycomb briquette combustion (a),
domestic chunk coal combustion (b), coal-fired power plant (c), and domestic coal combustion (d)
in 2016. The CFC-11 emissions from coal-fired power plant in January and February didn't have
specific data.
Shandong and Hebei Province were regarded as the hot source regions of CFC-
11 (Montzka et al., 2018; Rigby et al., 2019), from Figure 9a, Hebei and Shandong
provinces held high emission intensities of CFC-11 from coal combustion, with
average emission intensities of 0.23 and 0.03 kg km$^{-2}$, respectively. We also found that
the CFC-11 emission from coal combustion in Hebei and Shandong provinces peaked
in 2016 (Figures 9b−9c). They increased by 11.2% and 19.7%, compared with those
of 2013, then decreased by 15.2% and 8.6% in 2019, respectively. Based on the WRF-



FLEXPART modeling, we found that the CFC-11 emissions from coal combustion in
Shandong and Hebei provinces could impact South Korean areas (Figures 9d–9e).
After being emitted from coal combustion in the two provinces, CFC-11 could
contribute to the monitored atmospheric concentrations in January as 254~1062 ppt
within 410 km surrounding the emission source regions, higher than the observed
value of 249±13 ppt at Mount Tai in winter 2017 to spring 2018 (Huang et al., 2021),
this distance might not influence the monitoring station outside China. Specifically,
CFC-11 emissions from coal combustion in Hebei were simulated to contribute 51.8
ppt of the ambient CFC-11 at the Gosan station in South Korea. Similarly, the
contributions from Shandong were simulated as 17.6 ppt. At the Gosan station in
South Korea, the measured CFC-11 concentration was 233.2 ppt in January 2021
through the Advanced Global Atmospheric Gases Experiment (AGAGE, https://www-
air.larc.nasa.gov/missions/agage/). Our simulations suggested that CFC-11 emissions
from coal combustion in Hebei and Shandong contributed approximately 51.8 ppt and
17.6 ppt, respectively. They accounted for ~30% of the measured ambient value.
Although this suggested that regional coal combustion sources could slightly
influence background monitoring data in coastal East Asia, the contribution remained
much smaller than from global PAU-related emissions and must be interpreted
cautiously. Notably, the results here also exhibited uncertainties or shortages. Firstly,
the CFC-11 emission factors from chunk coal and honeycomb briquettes varied in a
large range, with the ratios of maximum to minimum values as 42 and 7 times, and
relative standard deviations of 61% and 81%, respectively. Secondly, the quick
variation of domestic coal consumption amount and structure had not been reflected
in the statistical yearbooks. The uncertainty of CFC-11 emission inventory from coal
combustion in this study was ±39.1% through 100000 Monte Carlo simulations.

**Figure 9.** The emission intensity (kg km$^{-2}$) of CFC-11 from coal combustion in 2016 (a), and the changes of CFC-11 from coal combustion in 2013, 2016, and 2019 in Hebei province (b) and Shandong province (c) in China. The distribution of simulated CFC-11 mass concentration contributed by coal combustion in January 2016 from Hebei (d) and Shandong (e) provinces with WRF-FLEXPART.

## 4. Data availability

The dataset presented is available at https://doi.org/10.6084/m9.figshare.28523063 (Niu et al., 2025). The activity data of coal combustion were from the China Energy Statistical Yearbook. Land use data was from https://data-starcloud.pcl.ac.cn/ (Gong et al., 2019, 2020). Population



distribution data were from https://hub.worldpop.org/doi/10.5258/SOTON/WP00674
(WorldPop and Center for International Earth Science Information Network). The POI
data of industrial were obtained from https://lbs.amap.com/. The CO emissions from
coal combustion in China were from http://meicmodel.org.cn/#firstPage
(Multiresolution Emission Inventory for China). The meteorological input data were
obtained and downloaded from https://rda.ucar.edu/ (National Centers for
Environmental Prediction Final Analysis).
**5. Conclusions**
There is currently no quantitative research on CFC-11 emissions from coal
combustion. This study addresses that gap by estimating CFC-11 emissions in China
from 2000 to 2021, based on coal consumption data and experimentally determined
emission factors. The measured CFC-11 EFs were 3.6, 3.2, and 0.025 mg kg$^{-1}$ for
domestic chunk coal, honeycomb briquettes, and coal-fired power plants, respectively.
During the study period, total CFC-11 emissions from coal combustion in China were
estimated at 233.5 t yr$^{-1}$. In Shandong and Hebei provinces, which have high levels of
coal consumption, CFC-11 emissions increased by approximately 74% during
2014~2017 compared to 2011~2012. At the Gosan monitoring station near mainland
China, emissions from Hebei and Shandong accounted for approximately 30% of the
average CFC-11 concentration in January 2016. Notably, China's clean heating and
coal-to-electricity policies also brought climate co-benefits, resulting in significant
reductions of $CO_2$-equivalent emissions by $2.2 \times 10^6$ tons and $3.4 \times 10^7$ tons,
respectively. This study provides quantitative evidence of CFC-11 emissions from
coal combustion, but the formation mechanisms of CFC-11 from coal combustion are
unclear and need further investigation.

**Author contributions:**
Zhenzhen Niu: Conceptualization, Experiments, Visualization, Writing; Shaofei Kong:
Conceptualization, Methodology, Supervision, Writing-review & editing. Qin Yan, Yi





Cheng, Huang Zheng, and Jian Wu: Experiments. Yao Hu, Xujing Qin, Haoyu Dong,
Weisi Jiang: Visualization. Yingying Yan, Wei Liu, Feng Ding, Yongqing Bai, and
Shihua Qi: Supervision.
**Competing interests:**
The contact author has declared that none of the authors has any competing interests.
**Financial Support:**
This work was supported by the Key Technologies Research and Development
Program (grant no. 2023YFC3709802), the Hubei Provincial Science Fund for
Distinguished Young Scholars (grant no. 2022CFA040), and the National Natural
Science Foundation of China (grant no. 42077202).





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
