# Peer review of "Estimation of CFC-11 emissions from coal combustion in China"

_Earth System Science Data, 2025_

## Author Comment (AC1)

Dear Reviewer:

Thank you for your help and valuable comments on our manuscript entitled "Estimation of CFC-11 emissions from coal combustion in China". These comments are very helpful for improving this manuscript and guiding our future research. We have viewed the comments carefully and have made corresponding corrections in the manuscript. Following lists the responses point by point.

**Responds to the reviewer's comments:**

The research "Estimation of CFC-11 emissions from coal combustion in China" detects CFC-11 emission factors from coal combustion in China and establishes a multi-year (2000~2021) emission inventory of CFC-11 from coal combustion in China. This study provides substantial evidence of CFC-11 emission from coal combustion and highlights the role of combustion emission under the background of reducing CFC-11 from production and use emission sources. I think the manuscript needs some minor revisions.

Response: Thanks for your positive opinions on this manuscript. We have addressed your comments point by point as follows. We really appreciated your suggestions, which helped us to improve this manuscript.

1. Line 22: "Results indicated that its annual…" was not clear.

Response: Thanks for this comment. "Results indicated that its annual emission averaged 233.5 t yr⁻¹" has been replaced by "Results indicated that CFC-11 annual emission from coal combustion in China averaged 233.5 t yr⁻¹".

2. Line 46~48: Give the specific GWP values of CFC-11 and $CO_2$.

Response: Thanks for this suggestion. "Over a 100-year time horizon, CFC-11 has a global warming potential (GWP) thousands of times greater than that of carbon dioxide ($CO_2$) (Chiodo & Polvani, 2022; Polvani et al., 2020), such as 7090 (Burkholder et al., 2022)." has been added in this part.

3. Line 91: "The second was a bottom-up method." is too simple.

Response: Thanks for this comment. The sentence has been changed to "The second method to estimate CFC-11 emissions was a bottom-up method.".

4. Line 103: What are the specific combustion sources?

Response: Thanks for this comment. "and combustion sources were always not included." has been changed as "and combustion sources (such as coal combustion, diesel combustion, etc.) were always not included.".

5. Line 123: in our combustion lab in Wuhan→ in the laboratory in Wuhan.

Response: Thanks for this suggestion. We have changed it according to your suggestion. It has been modified as "A total of 10 kinds of chunk coals and 11 kinds of honeycomb briquettes were collected and burned in the laboratory in Wuhan.".

6. Line 124~131: Add a map to show the location of these eight agricultural regions.

Response: Thanks for this suggestion. We have added figure as Figure S1 in the supporting information.

[Figure]

Figure S1 The location of eight agricultural regions of China

7. Line 223: Previous studies → Previous study.

Response: Thanks for this suggestion. We have changed it to "Previous study…".

8. Line 255: mg/kg → mg kg$^{-1}$.

Response: Thanks for this suggestion. We have changed it according to your suggestion.

9. Line 287~290: The comparation here is meaningless, the main topic here is that stationary combustion of biomass, trucks, vessels, and corn stover burning were also the CFC-11 emission sources. Rewritten this part.

Response: Thanks for this suggestion. This part has been rewritten as "The CFC-11 also detected from stationary combustion of biomass (0.01%) (Ye, 2018), heavy-duty diesel

trucks (0.05% or 0.2%) (Wang et al., 2020), vessels (0.07%) (Ye, 2018), and corn stover burning (0.3% or 0.2%) reported in the literature (Figure 1c) (Sun et al., 2018)."

10. Line 299: In this part, you only show the spatial emission in 4 years (Fig. 4), give other years' result in supporting information.

Response: Thanks for this suggestion. I have added figure (Figure S7 CFC-11 emission intensity of CFC-11 from domestic coal combustion in 2005~2020) in supporting information.

[Figure]

Figure S7 CFC-11 emission intensity of CFC-11 from domestic coal combustion in 2005~2020

11. Line 326: Label is not clear. Honeycomb → Honeycomb briquetted combustion; Power plant → Coa-fired power plant; Chunk coal → Chunk coal combustion

Response: Thanks for this suggestion. We have changed the label according to your suggestion.

[Figure]

Figure 3. The CFC-11 emissions (a~c) and mass percentages (d~f) from power plant, domestic chunk coal, and honeycomb combustion in Hebei, Shandong, and Sichuan provinces.

12. Line 343~345: Add a Figure to present the location of these provinces.

Response: Thanks for this suggestion. We have added figure (Figure S6) in supporting information to show the location of heating and non-heating areas in China.

[Figure]

Figure S6 The heating and non-heating areas in China

13. Line 352: Add markers to show the location of Hebei, Shandong, and Henan.

Response: Thanks for this suggestion. I have added markers in the Figure as follows:

[Figure]

Figure 4. CFC-11 emission intensity of CFC-11 from domestic coal combustion in 2005, 2010, 2015, and 2020

14. Line 390: "After 2025, the CFC-11 emissions from PAU in China was 0" adds reference.

Response: Thanks for this suggestion. The sentence has been changed to "After 2025, the CFC-11 emissions from PAU in China was 0 (Fang et al., 2018),".

15. Line 465: Label is not clear. Honeycomb → Honeycomb briquetted combustion; Power plant → Coal-fired power plant; Chunk coal → Chunk coal combustion

Response: Thanks for this suggestion. I have changed the label according to your suggestion.

[Figure]

Figure 7. $CO_2$-eq emissions for CFC-11 emitted from coal combustion in China in this study (a) and (b). The changes of $CO_2$-eq emissions from CFC-11, if coal was replaced by honeycomb (c) and if domestic coal was replaced by electricity produced in coal-fired power plant (d).

16. Line 487: 0.23 → 0.23 kg km$^{-2}$.

Response: Thanks for this suggestion. I have changed it (0.23→0.23 kg km$^{-2}$) according to your suggestion.

17. Line 495: What is the meaning of 410 km? Where is the influencing area within 410 km?

Response: Thanks for this comment. I have added the sentence in this part: CFC-11 emissions from coal combustion in the two provinces could lead to localized atmospheric concentrations of 254~1062 ppt within a 410 km radius of the source regions during January. These values exceed the observed CFC-11 concentration of 249 ± 13 ppt at Mount Tai during winter 2017 to spring 2018 (Huang et al., 2021). The influence within 410 km is unlikely to extend to monitoring stations outside China.

18. Line 538: for → from

19. Response: Thanks for this suggestion. We have replaced "for" with "from".

20. Line 547: tons → t

Response: Thanks for this suggestion. We have replaced "tons" with "t".

21. Line 764: (In Chinese) → China (In Chinese).

Response: Thanks for this suggestion. It has been modified as "China (In Chinese).".

22. Did the authors consider the CFC emissions from old refrigeration equipment, such as the process of scrapping?

Response: Thanks for this comment. According to Fang et al. (2018), sales of refrigerators using CFC-12 were completely in China banned by the end of 2007. CFC-11 is used in making foam insulation in freezers. The consumption and emission of foam agents are included under the foam-blowing sector as seen in Figure 2f.

23. How about the emissions of form utilization, e.g., in constructions?

Response: Thanks for this comment. In Figure 2, foam-blowing sector includes CFC-11 rigid polyurethane (PU) foam subsector, CFC-11, CFC-12 flexible polyurethane foam subsector, refrigerator foam subsector, and freezer foam subsector. But we didn't calculate the CFC-11 emission from constructions separately. In the future, we will consider calculating CFC-11 emission from all the possible emission sources.

---

## Author Comment (AC2)

Dear Reviewer:

Thank you for your help and valuable comments on our manuscript entitled "Estimation of CFC-11 emissions from coal combustion in China". These comments are very helpful for improving this manuscript and guiding our future research. We have viewed the comments carefully and have made corresponding corrections in the manuscript. Following lists the responses point by point.

This study obtained CFC-11 emission factors (EFs) for the combustion of typical Chinese domestic coal (chunk coal and honeycomb) through laboratory combustion experiments. EFs for coal-fired power plants were obtained through field sampling. Based on China's coal consumption data from 2000 to 2021, CFC-11 emission inventory for coal combustion in China was developed. Additionally, the study used Monte Carlo simulations to analyze the uncertainty in the emission inventory. The study systematically assessed the sources, emission amount, and trends of CFC-11 emissions from coal combustion in China. It improved the identification of non-conventional ODS sources in China, provided scientific basis for China's implementation of the Montreal Protocol and formulation of atmospheric pollution control policies. It is suggested that the manuscript be accepted after revising the following issues:

Response: Thanks for your positive opinions on this manuscript. We have addressed your comments point by point as follows. We appreciated your suggestions, which helped us to improve this manuscript.

1. Stove operation habits significantly influence pollutant emissions from domestic coal combustion. It is suggested to clarify whether the stove operation methods in the experiment align with the actual coal usage practices of rural residents in China.

Response: Thanks for this comment. We have carefully considered your suggestions. The experiment method has been adopted for many years in our group and was described in our previous papers (Yan et al., 2020; 2022). We wrote this section briefly before, now we accepted the suggestions and added the following sentences to the manuscript as follows:

To minimize the impact of ignition smoke, both honeycomb briquettes and chunk coal were lit from beneath pre-measured charcoal. An electric oven was used to ignite

the charcoal, allowing it to burn until visible smoke dissipated. The combustion state was controlled by adjusting the stove's bottom air door: fully open for flaming and closed for smoldering. This method replicated the actual burning practices observed in rural China (Yan et al., 2020; 2022).

Yan, Q., Kong, S., Yan, Y., Liu, H., Wang, W., Chen, K., et al. (2020). Emission and simulation of primary fine and submicron particles and water-soluble ions from domestic coal combustion in China. *Atmospheric Environment*, *224*, 117308. https://doi.org/10.1016/j.atmosenv.2020.117308

Yan, Q., Kong, S., Yan, Y., Liu, X., Zheng, S., Qin, S., et al. (2022). Emission and spatialized health risks for trace elements from domestic coal burning in China. *Environment International*, *158*, 107001. https://doi.org/10.1016/j.envint.2021.107001

2. The paper refers to that the EFs for CFC-11 exhibits significant variability (chunk coal: 0.3–12.7 mg/kg). It is suggested to include an analysis of the reasons for these fluctuations, such as the impact of halogen content in the coal.

Response: Thanks for this suggestion. As described in Section 3.1, from previous studies, CFC-11 includes chloride (Cl) and fluoride (F), the formation of CFC-11 needs the participation of Cl and F. F and Cl were widely distributed in China's coal (Jin et al., 2025; Yang et al., 2017). Former studies indicated that F content was 20~300 mg $kg^{-1}$ from coals in the North China Plane and Northwest China, lower than the Southwest China (50~3000 mg $kg^{-1}$) (Luo et al., 2004). The F content in China's coal was 11~3575 mg $kg^{-1}$, averaged as 130 mg $kg^{-1}$ (Yang et al., 2017). The chlorine content of bituminous coal was 252.5 mg $kg^{-1}$ in China (Jin et al., 2025). Chen et al. (2010) collected 305 kinds of coal samples all around China and analysis the Cl content, results indicated that the Cl content in different provinces were also different, which were 13.2~2815 µg $g^{-1}$. We agree with you and infer that halogen content in the coal may influence the EFs of CFC-11 from coal combustion. Unfortunately, we discovered that the combustion of coal could produce CFC-11 by accident, so we didn't analyze

the content of halogen in the coal. In the future, the formation and emission mechanisms of CFC-11 during coal combustion and the influencing factors need deep research.

Chen, L. (2010). Study on environmental geochemistry of Chlorine in Chinese coals. Nanchang University.

Jin, W., Yan, Y., Qiu, X., Peng, L., Li, Z., & Tang, Y. (2025). Characterizing full-phase chlorine species emissions from domestic coal combustion in China: Implications for significant impacts on air pollution and ozone-layer depletion. *Environmental Pollution*, 372, 126043. https://doi.org/10.1016/j.envpol.2025.126043

Yang, N., Tang, S., Zhang, S., Huang, W., Chen, P., Chen, Y., et al. (2017). Fluorine in Chinese coal: A review of distribution, abundance, modes of occurrence, genetic factors and environmental effects. *Minerals*, *7*, 219. https://doi.org/10.3390/min7110219

3.   The paper uses Monte Carlo simulation for uncertainty analysis. It is suggested to clarify the sources of the coefficient of variation for activity data and EFs in the simulation.

Response: Thanks for this comment. I have added the detailed information as follows: The uncertainty of CFC-11 emission inventory from coal combustion in 2021 was ±50.2% through 100000 Monte Carlo simulations with a 95% coincidence interval. In this study, the coefficients of variation (CV, the standard deviation divided by the mean) for coal consumption in power plant was assumed as 5%, and for domestic coal consumption it was 20% (Zhao et al., 2011). The uncertainty for EFs were calculated according to the EFs from experiment in this study (Table S2).

Table S2 The emission factors (mg $kg^{-1}$) of domestic coal combustion used for estimating domestic CFC-11 emissions.

| Types | Sources | CFC-11 |
|---|---|---|
| Chunk coal | Northeast Plain | 5.5±0.4 |
| | Arid and semi-arid regions of north China | 5.6±2.1 |
| | Loess Plateau | 3.6±3.0 |
| | North China plain | 2.1±0.3 |

| | | |
|---|---|---|
| | Yangtze Plain | 2.5±3.7 |
| | Sichuan Basin | 3.4±0.8 |
| | Yunnan-Guizhou Plateau | 1.6±3.6 |
| | Tibet Plateau | 2.2±1.7 |
| | South China | 1.6±3.3 |
| | Northeast Plain | 3.3±9.3 |
| | Arid and semi-arid regions of north China | 1.5±1.5 |
| | Loess Plateau | 3.3±10.9 |
| | North China plain | 3.8±9.9 |
| Honeycomb briquette | Yangtze Plain | 3.2±12.7 |
| | Sichuan Basin | 3.1±10.9 |
| | Yunnan-Guizhou Plateau | 1.5±1.7 |
| | Tibet Plateau | 3.3±9.3 |
| | South China | 4.7±10.9 |
| Coal | Power plant | 0.02±0.004 |

Zhao, Y., Nielsen, C. P., Lei, Y., McElroy, M. B., and Hao, J. (2011). Quantifying the uncertainties of a bottom-up emission inventory of anthropogenic atmospheric pollutants in China. *Atmospheric Chemistry and Physics*, *11*, 2295–2308. https://doi.org/10.5194/acp-11-2295-2011

4.   The paper uses the CO tracer method to calculate CFC-11 emissions. It is suggested to supplement relevant details about this method in Chapter 2.

Response: Thanks for this suggestion. I have added the method in Section 2.3 as follows:

When using CO as a tracer to calculate the CFC-11 emissions, the method was as follows (Palmer et al., 2003):

$$E_{CFC-11} = E_{CO} \times \frac{\Delta CFC-11}{\Delta CO} \times \frac{M_{CFC-11}}{M_{CO}} \quad\quad (4)$$

In which $E_{CFC-11}$ was the CFC-11 emissions, t; $E_{CO}$ was the CO emissions, t; $\frac{\Delta CFC-11}{\Delta CO}$ was the slope of the linear correlation between ΔCFC-11 and ΔCO; $M_{CFC-11}$ and $M_{CO}$ were the molecular weights of CFC-11 and CO.

Palmer, P. I., Jacob, D. J., Mickley, L. J., Blake, D. R., Sachse, G. W., Fuelberg, H. E., et al. (2003). Eastern Asian emissions of anthropogenic halocarbons deduced from aircraft concentration data. *Journal of Geophysical Research: Atmospheres*, *108*(D24), 2003JD003591. https://doi.org/10.1029/2003JD003591

5. In Fig. 3(e) and (f), there is a significant difference in the emission quantities and their proportions for different coal types. Please confirm if the data in the images are accurate.

Response: Thanks for this comment. I have checked the data and put the right Figure in the draft. The right Figure is as follows:

[Figure]

Figure 3. The CFC-11 emissions (a~c) and mass percentages (d~f) from power plant, domestic chunk coal, and honeycomb combustion in Hebei, Shandong, and Sichuan provinces.